# Proximal femoral nail antirotation versus InterTan nail for the treatment of intertrochanteric fractures: A systematic review and meta-analysis

**Chang-sheng Liao** [1,2‡], **Fang-zheng He** [1,2‡], **Xi-yong Li** [1,2], **Peng-fei Han** [1]*

1 Department of Orthopaedics, Heping Hospital Affiliated to Changzhi Medical College, Changzhi, P.R. China, 2 Department of Graduate School, Graduate Student Department of Changzhi Medical College, Changzhi, P.R. China

‡ CSL and FZH are co-first authors on this work.
* 18003551149@163.com

**Data Availability Statement:** All relevant data are within the manuscript and its Supporting Information files.

## Abstract

### Purpose

This meta-analysis compared the efficacy and safety of Proximal Femoral Nail Antirotation (PFNA) and InterTan Nail in the treatment of intertrochanteric fractures. Given the high incidence of femoral intertrochanteric fractures in the elderly population and its impact on quality of life, choosing the most effective and safest surgical option is crucial. PFNA and InterTan are currently two commonly used techniques, but there is a lack of systematic evaluation comparing their safety and effectiveness. This study aims to fill this knowledge gap through Meta-analysis, providing clinicians with evidence-based treatment recommendations.

### Materials and methods

A computer search was used to search for published literature on PFNA and InterTan in the treatment of intertrochanteric fractures in PubMed (Medline), Web of Science, Embase, Cochrane Library (CENTRAL), Cinahl, CBM, and CNKI.A total of 853 related literatures were retrieved, and 15 literatures were finally included. Newcastle-Ottawa-Scale and Cochrane systematic review methodologies were used to assess the quality of the literature. Meta-analysis was performed using Review Manager 5.4 software, following data extraction.

### Results

The comparison found that during the surgical treatment of intertrochanteric fractures, the operation time, fluoroscopy time, and blood loss in the PFNA group were significantly shorter than those in the InterTan group, and the difference was statistically significant. In terms of postoperative complication rates, the InterTan group had a significant advantage over the PFNA group. Shaft fracture, varus collapse, cut out, screw migration, and pain of hip and thigh were the most likely to occur in the PFNA group, and the differences were all

**Funding:** The author(s) received no specific funding for this work.

**Competing interests:** The authors have declared that no competing interests exist.

**Abbreviations:** PFNA, Proximal Femoral Nail Antirotation; TAD, Tip-Apex Distance; HHS, Harris Hip Score.

statistically significant. In terms of postoperative efficacy, the results of the PFNA group and the InterTan group were comparable, and there was no significant differences.

## Conclusions

When selecting surgical techniques for the treatment of femoral intertrochanteric fractures, it is necessary to conduct individualized assessments based on the patient's overall health status, surgical tolerance, and post-operative recovery needs. For patients who cannot tolerate long-term surgery or are in poor physical condition, PFNA may be more appropriate. While for patients who can tolerate long-term surgery or have more complex conditions, InterTan may be more suitable.

## 1. Introduction

Femoral intertrochanteric fractures, also known as femoral intertrochanteric fractures, occur below the hip joint capsule and above the level of the lesser trochanter [1]. These fractures belong to the category of extracapsular fractures, and due to the dense vascular distribution at the fracture site, problems such as non-union of the wound or avascular necrosis of the femoral head occur less frequently. However, the use of conventional conservative treatment methods may lead to issues such as malformed healing of the joint and unequal lengths of the lower limbs [2]. Therefore, unless there are clear contraindications, surgical fixation has become the standard approach for treating femoral intertrochanteric fractures. In past practices, conservative treatment often carried complications associated with prolonged bed rest and immobilization, such as bedsores, disuse osteoporosis, venous thrombosis, and pneumonia [3]. To improve treatment outcomes, intramedullary fixation has gradually become the preferred method for treating femoral intertrochanteric fractures, especially unstable ones. Compared to extramedullary fixation, the intramedullary fixation system, with its screws positioned within the medullary cavity, better aligns with the physiological load-bearing line, effectively reducing stress at the screw-rod interface, thereby enhancing stability and facilitating load transfer [4]. Additionally, the closed reduction technique used in intramedullary fixation reduces tissue damage and bleeding, without directly interfering with the blood supply to the fracture ends, facilitating early recovery and functional restoration for patients. Among the intramedullary fixation systems, PFNA and InterTan are two commonly used techniques. Although numerous studies have explored the application of these two techniques in the treatment of femoral intertrochanteric fractures, there are significant differences in their conclusions. Some studies indicate that the InterTan system exhibits superior biomechanical properties and clinical outcomes [5], while others report that InterTan does not significantly improve patients' functional recovery [6]. Given these controversies, this study aims to systematically evaluate the clinical outcomes of PFNA and InterTan in the treatment of femoral intertrochanteric fractures using meta-analysis methods, in order to provide more objective and comprehensive evidence to guide clinicians' decision-making.

## 2. Materials and methods

This study strictly follows the PRISMA (Preferred Reporting Items for Systematic Reviews and Meta-Analyses) reporting system to ensure transparency and reproducibility of the research. Additionally, this study has been registered on PROSPERO (International Prospective Register

of Systematic Reviews) with the registration number of CRD42024534873 to further enhance the transparency and credibility of the research. On this premise, we have adopted the following rigorous methodologies for literature retrieval, screening, data extraction, and quality assessment.

## 2.1. Inclusion and exclusion criteria

The subjects included in this study are patients from published clinical controlled studies who have been diagnosed with femoral intertrochanteric fractures through a comprehensive evaluation of medical history, physical examination, and radiological imaging, and have been determined to require surgical intervention. Documents such as non-clinical controlled studies, case reports, reviews, letters, conference paper abstracts, and duplicate reports are excluded from the study. The primary intervention measures are the two techniques of PFNA and Inter-Tan. The outcome indicators include objective metrics like surgical duration, fluoroscopy time, blood loss, and length of hospital stay, as well as comprehensive indicators such as Harris Hip Score, reduction quality, Tip-Apex Distance (TAD), bone union status, and occurrence of complications.

## 2.2. Search strategy

This study conducted an extensive literature search in multiple databases, including PubMed, Web of Science, Embase, Cochrane Central Register of Controlled Trials, Cinahl, Medline, Cochrane Library, CBM, and CNKI. Relevant journal catalogues and references were also reviewed. No specific restrictions were placed on the sample size or age range of participants, and there were no language limitations for inclusion criteria. The search keywords encompassed "intramedullary nail", "Proximal femoral nail antirotation", "Intertrochanteric femoral fracture", "intertan", and "PFNA". The search strategy was formulated as follows: ((((intramedullary nail) OR (intertan) OR (PFNA)) OR (intramedullary hip screw)) OR (Proximal femoral nail antirotation)) AND ((trochanteric femoral fracture) OR (Intertrochanteric femoral fracture)).Additionally, the literature search for this study was limited to the time period from January 2002 to June 2023, to ensure that the retrieved literature aligns with current research backgrounds and trends.

## 3. Data screening and extraction

### 3.1 Screening process

We implemented rigorous initial and secondary screening processes to ensure that only studies meeting our analytical criteria were included. Initial screening was based on the research topic, type (such as RCTs, cohort studies, etc.), characteristics of participants, and the relevance of intervention measures to outcome indicators. Secondary screening involved reading the full texts, assessing the research design, data quality, sample size, and statistical methods. Exclusion criteria included studies with serious methodological flaws, incomplete data, or data that could not be accessed. Zotero software was used to manage the literature, avoiding duplicate records and retaining only unique ones.

### 3.2 Bias control

Multiple rounds of screening were conducted, with each round independently performed by different researchers to reduce subjective bias. Independent reviews were conducted for inclusion and exclusion decisions, and consensus was reached through team discussions.

### 3.3 Data extraction

Key variables were extracted, including patient information, surgical details, effectiveness assessments, and postoperative complications. A combination of manual extraction and data extraction forms was used to ensure the comprehensiveness and consistency of information. Emphasis was placed on the accuracy and reliability of data, and ambiguous data were verified. Missing data were handled by attempting to supplement or adopting imputation methods, and sensitivity analysis was conducted. If there were excessive missing data, the study was considered for exclusion and noted in the results.

### 3.4 Quality assessment

Two researchers independently extracted data using the designed search strategy, and any differences were resolved through discussion until a consensus was reached or the quality of the literature was jointly evaluated with a third researcher. Strictly adhere to the Cochrane risk of bias assessment criteria, which include the following: i) ensuring the incorporation of the randomization principle into the experimental design; ii) maintaining adherence to the double-blind principle among both participants and researchers; iii) verifying the accuracy and reliability of experimental statistics; iv) implementing the allocation concealment method during the experiment; v) assessing whether the selective reporting method has been employed in the experimental process; and vi) considering any other potential bias factors that may influence the outcome. Simultaneously, the quality of the literature was evaluated using the Newcastle-Ottawa-Scale (NOS), which comprises three dimensions encompassing eight items: four items for the selection of the study population, one item for comparability between groups, and three items related to outcomes measurement. Except for the comparability item, which can score a maximum of two points, each other item can score a maximum of one point, resulting in a total score ranging from zero to nine points. The higher the overall score, the higher the quality of the study. If a study includes multiple cohorts, we will assign scores to each cohort individually. Among the outcomes indicators, the follow-up time was defined as $\geq$one year, and the loss-to-follow-up rate was $\leq$15%. Based on the NOS scores, we categorized the studies into three quality levels: low ($<$5 points), medium (5 to $<$8 points), and high (8 to 9 points) [7].

### 3.5 Statistical analysis

We conducted a meta-analysis of the extracted data using Review Manager 5.4. Mean differences (MD) and 95% confidence intervals (CI) were utilized for continuous variables, while odds ratios (OR) and 95% CI were applied to dichotomous variables. Heterogeneity was assessed using the $I^2$ statistic: if $I^2 <$ 50%, indicating low heterogeneity among studies, a fixed effects model was employed. In cases of significant heterogeneity ($I^2 >$ 50%), a random effects model was preferred to account for the variability across studies. At this juncture, it is imperative to assess publication bias, conduct a sensitivity analysis, and investigate potential sources of heterogeneity. A P-value less than 0.05 was deemed statistically significant for our analysis. The report was prepared following the PRISMA guideline.

## 4. Result

### 4.1. Included literature features

As a result of our search strategy, a total of 853 literature articles were retrieved. After screening the titles and abstracts, we excluded 818 non-controlled studies, duplicates, and articles unrelated to our research objectives, initially filtering out 34 relevant articles. Upon further reading of the full texts and strict adherence to the inclusion and exclusion criteria, we

ultimately included 15 articles. From each eligible study, we extracted data including author details, study nature, number of cases, mean age of patients, gender, treatment of intertrochanteric fractures, intraoperative indicators, postoperative efficacy indicators, and postoperative complications. The extracted data exhibited comparable characteristics across the included studies, as indicated by statistical tests with P-values greater than 0.05.The process and results of the literature screening are graphically presented in Fig 1. Meanwhile, S1 Table summarizes the characteristics of the 15 included studies [8–22].

## 4.2. Quality assessment of included literature

After rigorous screening and evaluation, a total of 15 articles were ultimately included for comprehensive analysis. Among them, 2 were prospective studies, and the remaining 13 were retrospective studies. It is worth noting that, although randomized controlled trials (RCTs) are often considered the gold standard for evaluating treatment effects and were not excluded during the initial stages of our research, no RCTs meeting the study objectives and criteria were identified among the finally included articles. Subsequently, we used the Newcastle-Ottawa Scale (NOS) to assess the quality of the literature. The results showed that 4 articles received a score of 8 and were designated as high-quality; 7 articles received a score of 7, 3 articles received a score of 6, and 1 article received a score of 5. These 11 articles were considered to be of moderate quality. Although the number of included articles was limited and may have introduced some bias, the overall quality was deemed to be moderate.

## 4.3. Outcomes

**4.3.1 Comparison of intraoperative indicators.** Intraoperative indicators encompassed Duration of surgery, Fluoroscopy time, and Blood loss. A total of 11 studies compared the surgical duration between PFNA and InterTan. The heterogeneity test ($I^2$ = 98%, P < 0.001) revealed significant heterogeneity among the studies, thus necessitating the application of a random effects model. In the treatment of intertrochanteric fractures of the femur, the results indicated that the PFNA group required a shorter operation time than the InterTan group, with a statistically significant difference [95%CI (-13.70, -3.12), P = 0.002]. A total of six studies compared Fluoroscopy time, with the heterogeneity test showing $I^2$ = 100%, indicating significant heterogeneity among the studies. Consequently, a random effects model was also employed. The results confirmed that the Fluoroscopy time of the PFNA group was shorter than that of the InterTan group in the treatment of intertrochanteric fractures, with a statistically significant difference [95%CI (-105.58, -27.32), P < 0.001]. Additionally, 10 studies compared blood loss between PFNA and InterTan. Heterogeneity tests demonstrated significant heterogeneity among the studies ($I^2$ = 96%, P < 0.001), necessitating the use of a random effects model. Eight of the studies reported lower mean blood loss in the PFNA group, while two studies favored InterTan. Overall, the results indicated that blood loss was less in the PFNA group compared to the InterTan group, with a statistically significant difference [95% CI (-25.95, -8.30), P < 0.001] (Fig 2).

**4.3.2 Comparison of postoperative efficacy indicators.** Postoperative efficacy indicators included Hospital stay, HHS, Good reduction quality, TAD and Union of bone. Through heterogeneity analysis, we found that Hospital stay ($I^2$ = 92.0%, P<0.001), HHS ($I^2$ = 77.0%, P<0.001), TAD ($I^2$ = 98.0%, P<0.001), there was significant heterogeneity between studies, so a random-effects model was used for classification. Good reduction quality ($I^2$ = 0%, P = 0.87) and Union of bone ($I^2$ = 0%, P = 0.42) showed no significant heterogeneity between studies and was therefore classified using a fixed effects model. The results showed that the two groups had similar postoperative efficacy indicators, including Hospital stay [95%CI (-1.10, 1.44),

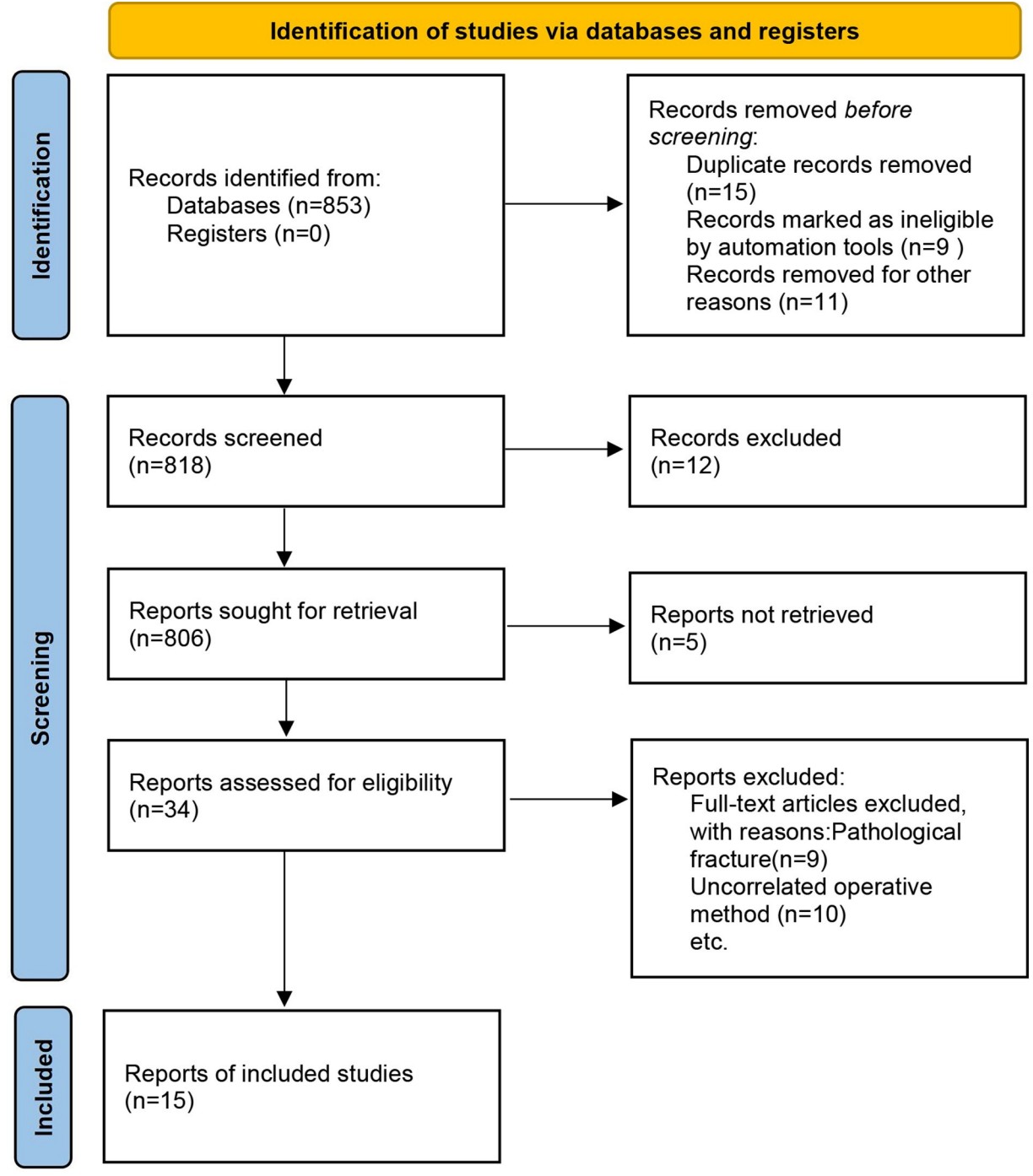

**Fig 1. Flowchart of literature screening.**

P = 0.79], HHS [95%CI (-2.16, 0.50), P = 0.22], Good reduction quality [95%CI(0.51, 1.86), P = 0.92], TAD [95%CI(-1.11, 1.30), P = 0.88], Union of bone [95%CI(-0.01, 0.72), P = 0.06], the difference was not statistically significant (Fig 3).

**4.3.3 Comparison of postoperative complications.** 10 articles compared postoperative complications of PFNA and InterTan, and we divided them into 20 subgroups according to

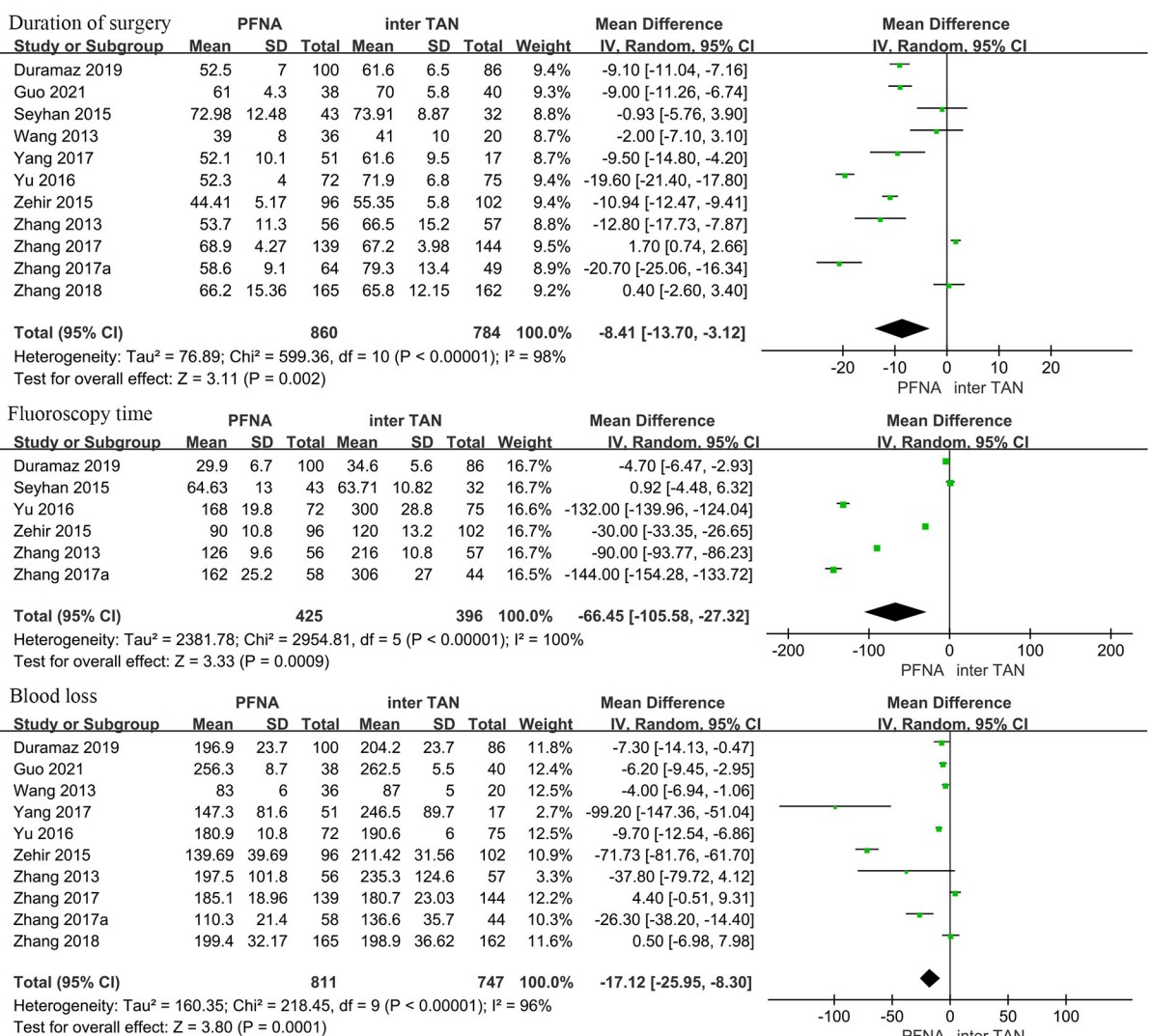

**Fig 2. Meta-analysis of intraoperative indicators for the treatment of intertrochanteric fractures with PFNA and InterTan.**

the type of complications. The overall heterogeneity test ($I^2$ = 7%, P = 0.35) indicated that there was no considerable heterogeneity between studies, therefore the fixed-effects model was used for classification. For certain subgroups, meta-analysis was not feasible due to insufficient data from three groups. In the heterogeneity test, the $I^2$ value for the remaining subgroups was below 50%, suggesting the absence of significant heterogeneity among the studies. Therefore, the fixed-effects model was utilized for classification. The consequences confirmed that the overall postoperative complication in the PFNA group was once greater than that in the InterTan group in the cure of intertrochanteric fractures, and the distinction used to be statistically significant [95% CI (2.08, 3.47), P<0.001]. In the subgroup, the probability of Shaft fracture, Varus collapse, Cut out, screw migration and Pain of hip and thigh in the PFNA group was higher than that in the InterTan group, and the differences were statistically significant, the values were [95%CI (2.39, 12.35), P<0.001], [95%CI (2.29, 26.53), P = 0.001], [95%CI (2.82, 13.51), P<0.001], [95%CI (1.70, 17.02), P = 0.004] and [95% CI (1.72, 4.93), P<0.001] (Fig 4).

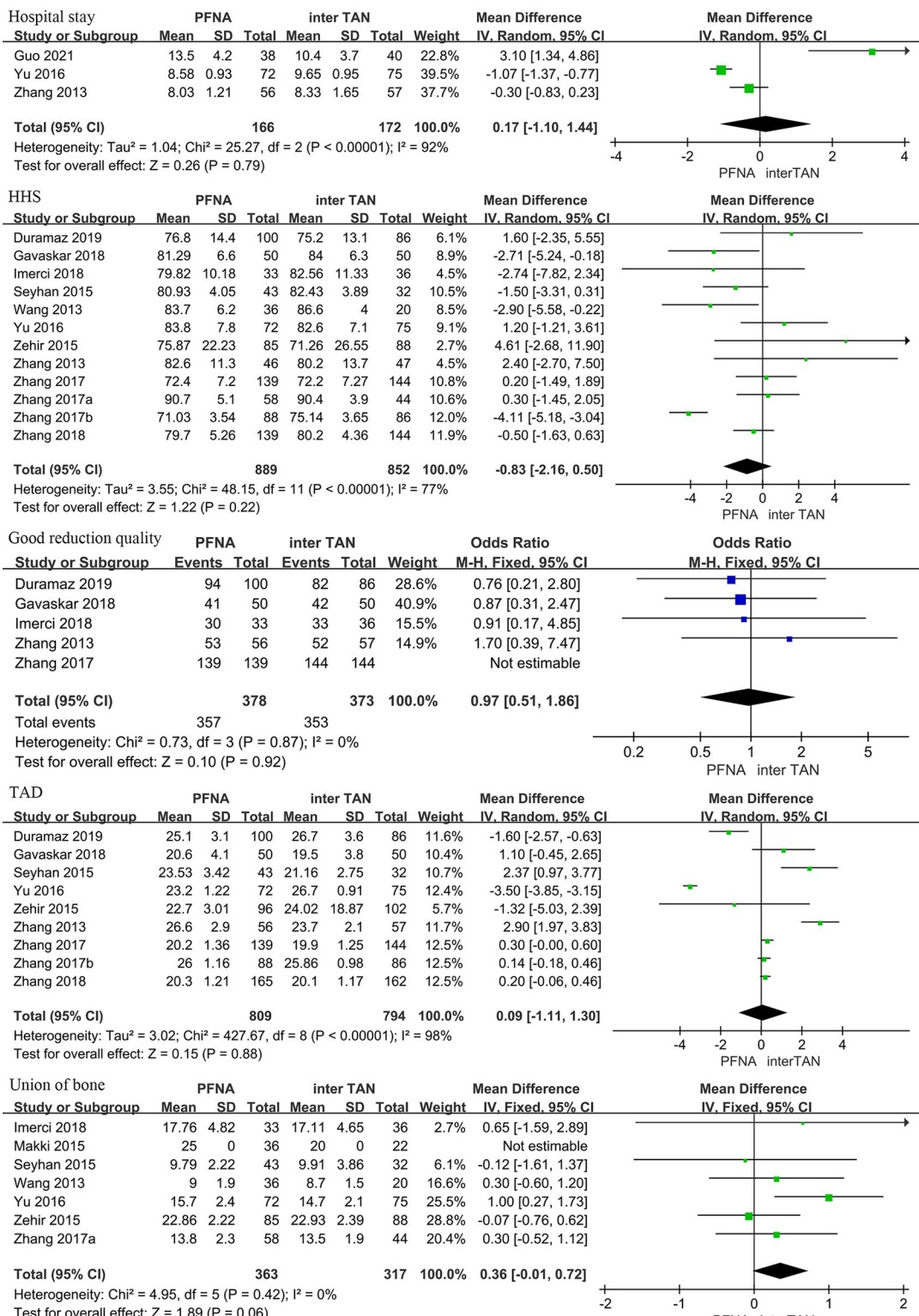

**Fig 3. Meta-analysis of postoperative outcomes for the treatment of intertrochanteric fractures with PFNA and InterTan.**

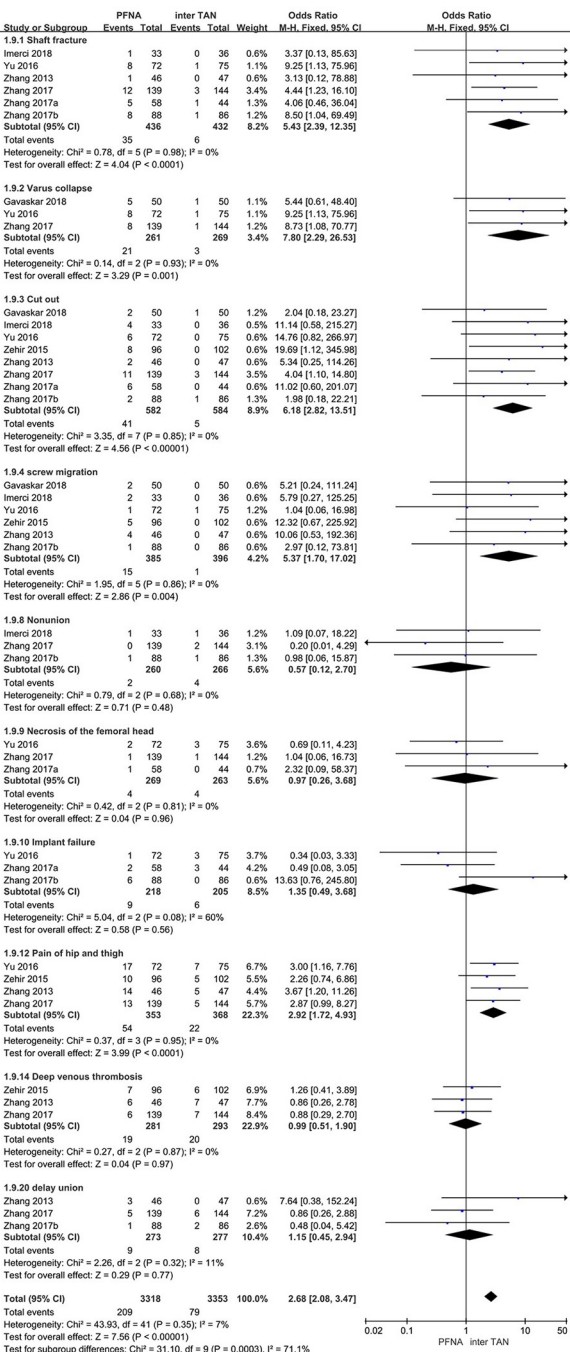

**Fig 4. Meta-analysis of postoperative complications for the treatment of intertrochanteric fractures with PFNA and InterTan.**

## 4.4. Publication bias and sensitivity analysis

We conducted publication bias and sensitivity analyses using Review Manager 5.4 software on nine outcome indicators: Duration of surgery, Fluoroscopy time, Blood loss, Hospital stay, Harris Hip Score (HHS), Good reduction quality, Tip-apex distance (TAD), Union of bone, and Complications associated with intertrochanteric fracture surgery. The results indicate that

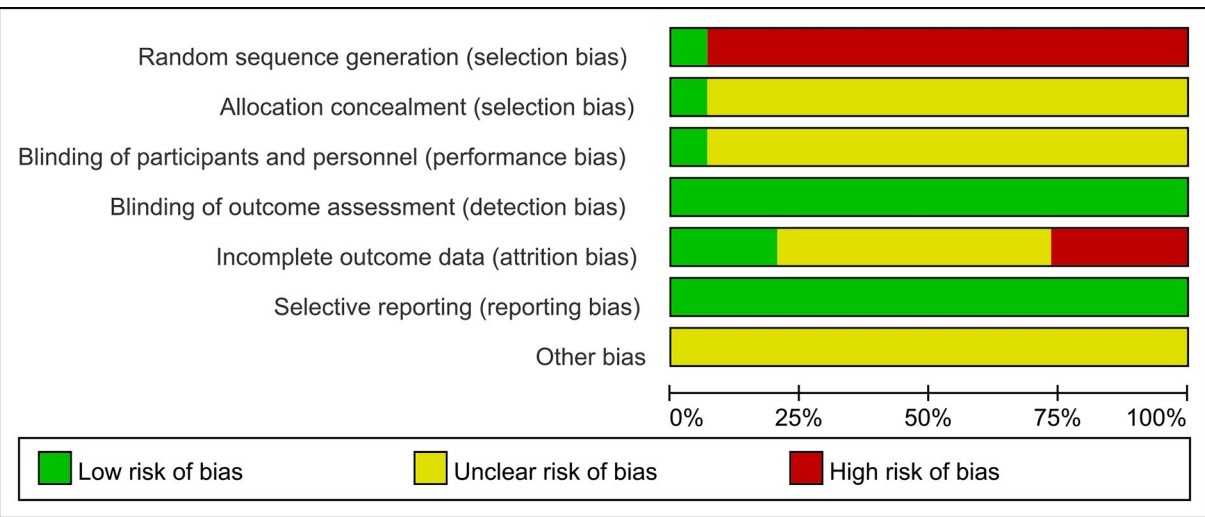

**Fig 5. Funnel plot analysis of publication bias and sensitivity analysis for various outcome indicators in surgical treatment of intertrochanteric fractures of the femur.**

the funnel plots are generally symmetrical, indicating the absence of significant publication bias and suggesting that the data are stable and reliable (Figs 5–9).

## 5. Discussion

Intertrochanteric fractures of the femur, a common type of hip fractures in clinical practice, primarily occur between the base of the femoral neck and the lesser trochanter, often caused by falls or indirect torsion between the femoral trochanters. They are closely associated with osteoporosis and are more common in the elderly [23]. Compared to the femoral neck below the femoral head, the bone in this fracture area is mainly composed of cancellous bone, lacks a capsule, but is rich in blood vessels, providing conditions for a good healing environment. The greater trochanter serves as the attachment point for the primary hip abductor (gluteus medius), while the lesser trochanter is the attachment point for the primary hip flexor (iliopsoas). The calcar femorale is an extension of the posterior medial cortex of the femoral body into the cancellous bone. From a biomechanical perspective, the calcar femorale and the compressive trabeculae of the inferomedial cortex of the femoral neck jointly bear the eccentric load on the femoral head, forming compressive stress and bending moment [24]. These anatomical features have a significant impact on the choice of treatment options. As age increases, the incidence of intertrochanteric fractures of the femur gradually rises [25]. Some elderly patients often suffer from chronic diseases such as diabetes and hypertension, which not only complicates the condition but also poses more challenges to treatment. The prognosis is generally poor, severely affecting the quality of life of patients.

Currently, intramedullary fixation surgery is the primary choice for the treatment of intertrochanteric fractures of the femur, with PFNA and InterTan being two commonly used treatment options [26]. Some scholars prefer PFNA, believing that it has significant efficacy in the treatment of intertrochanteric fractures of the femur, which is consistent with the views of Duramaz et al. also pointed out that due to the need for more frequent intraoperative fluoroscopy during the InterTan surgical procedure, PFNA can be considered a preferred option in some cases [8]. However, some scholars have also proposed that PFNA may have certain limitations in dealing with complex intertrochanteric fractures, especially when the lateral wall is

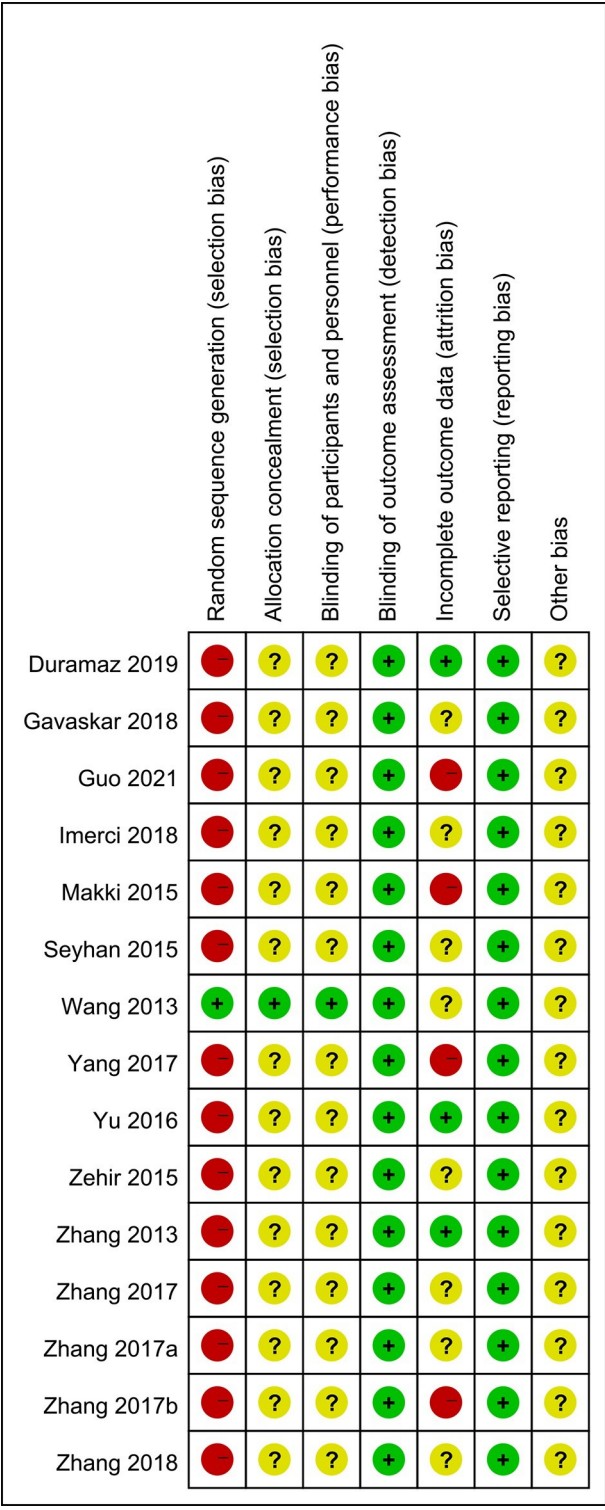

**Fig 6. Funnel plot analysis of publication bias and sensitivity analysis for various outcome indicators in surgical treatment of intertrochanteric fractures of the femur.**

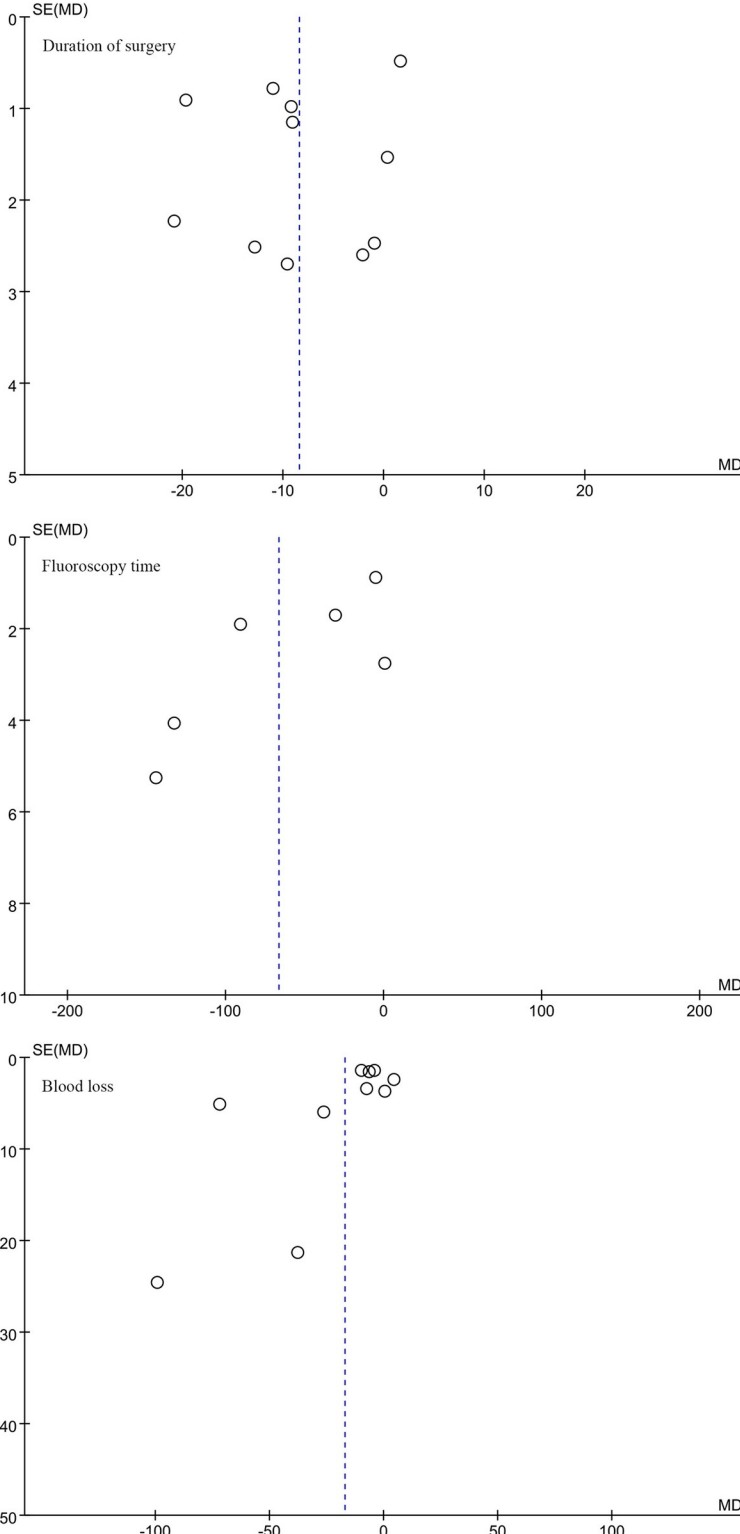

**Fig 7. Funnel plot analysis of publication bias and sensitivity analysis for various outcome indicators in surgical treatment of intertrochanteric fractures of the femur.**

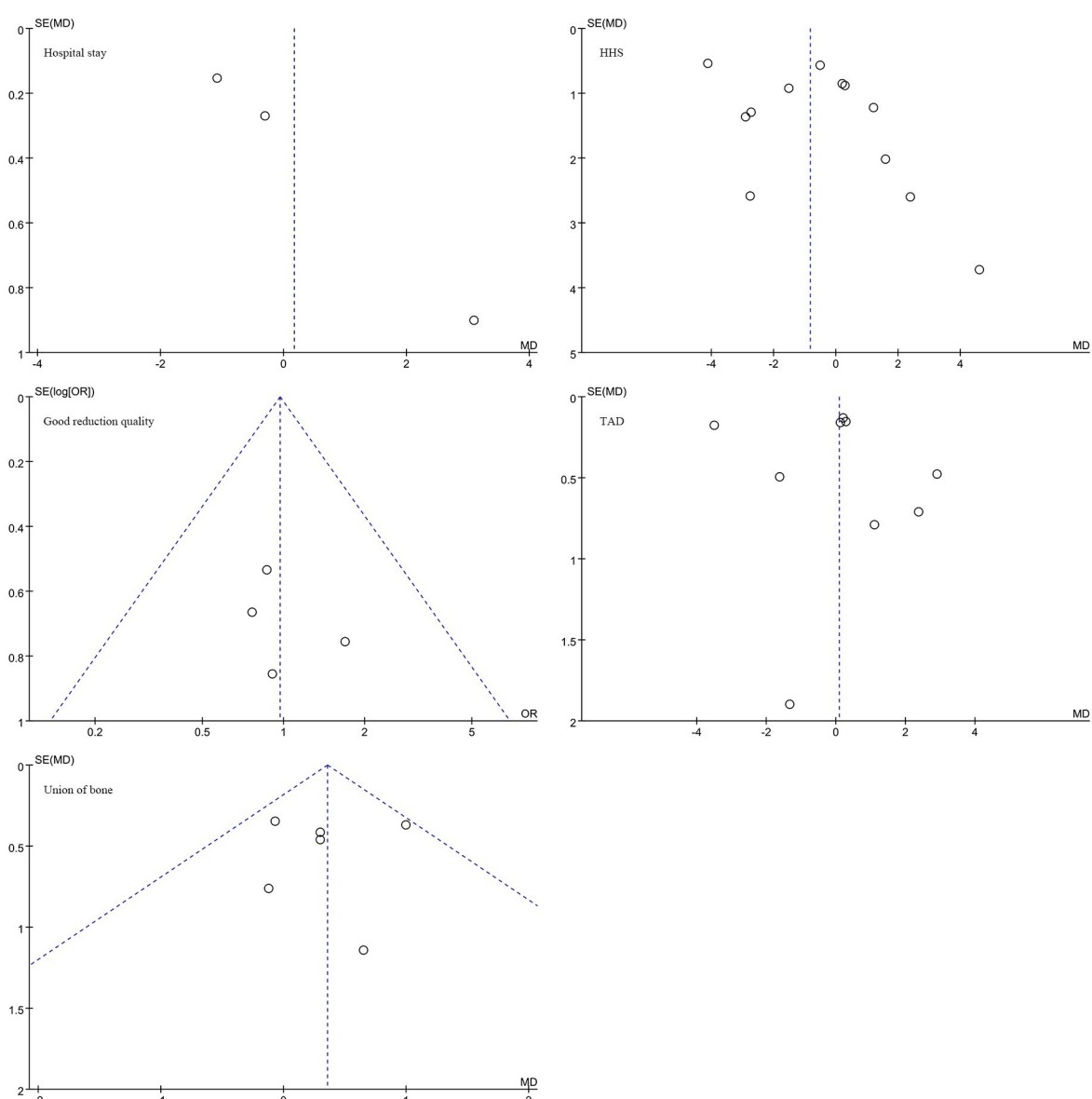

**Fig 8. Funnel plot analysis of publication bias and sensitivity analysis for various outcome indicators in surgical treatment of intertrochanteric fractures of the femur.**

incomplete. Since PFNA belongs to the classic sliding compression fixation method, incomplete lateral walls may lead to complications such as nail backout [27]. In contrast, the Intertan intramedullary fixation adopts a static locking fixation method, which not only improves antirotation stability and axial compression ability but also effectively avoids sliding phenomena when treating fractures with incomplete lateral walls, reducing the incidence of postoperative femoral head rotation and thus reconstructing the function of the lateral wall to a certain extent [28]. Yu et al. further emphasized that when treating complex intertrochanteric fractures with incomplete lateral walls, the complication rate of InterTan is relatively low. Nevertheless, it is worth noting that due to its locking fixation method, InterTan cannot achieve dynamic sliding compression at the fracture site, which may lead to a relatively prolonged fracture healing time [29]. Therefore, when choosing a surgical method, doctors need to

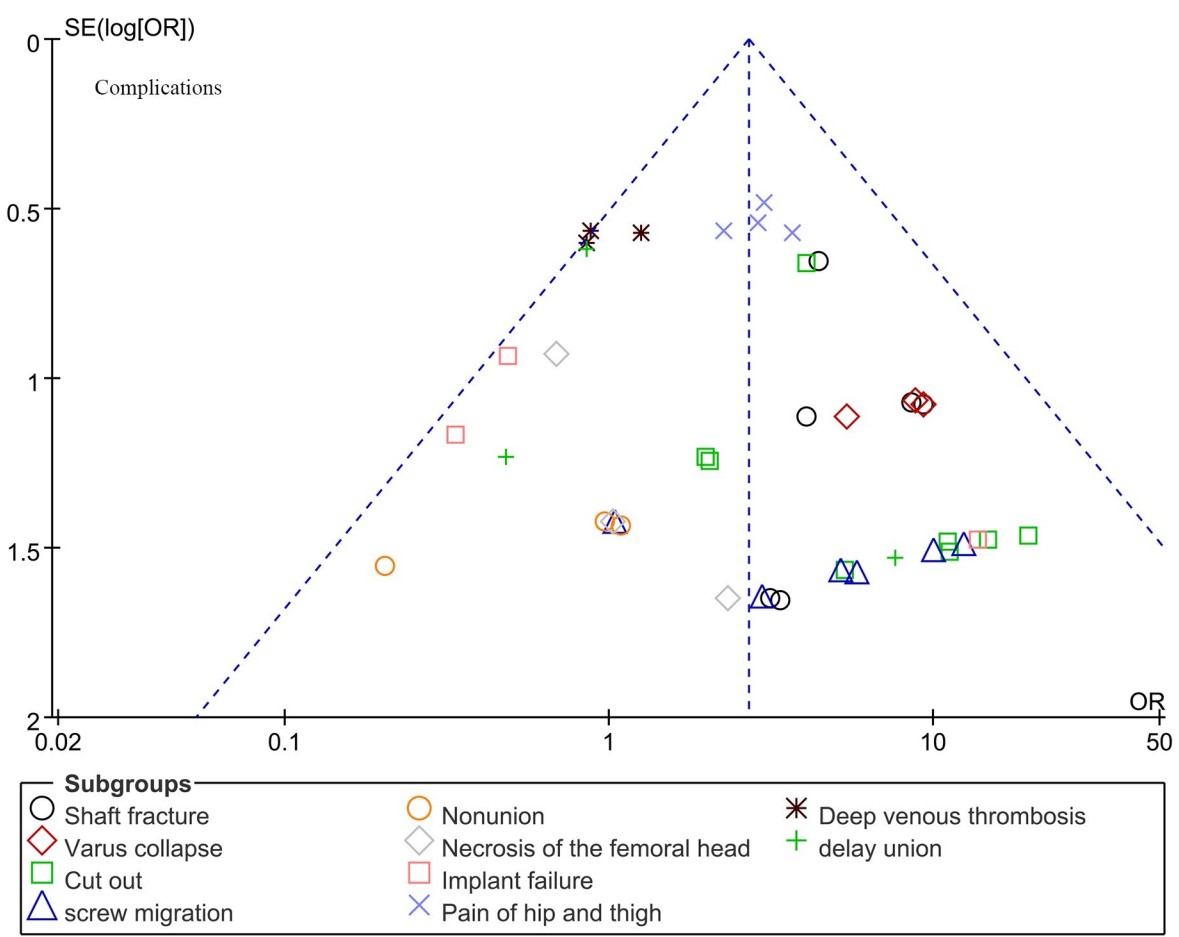

**Fig 9. . Funnel plot analysis of publication bias and sensitivity analysis for various outcome indicators in surgical treatment of intertrochanteric fractures of the femur.**

comprehensively consider various factors, including the specific type of fracture and the individual situation of the patient, to determine the most suitable treatment plan for the patient.

The present meta-analysis aims to compare the therapeutic effects of PFNA and InterTan in the treatment of femoral intertrochanteric fractures. Nine outcome indicators were selected for evaluation, including surgical duration, fluoroscopy time, blood loss, length of hospital stay, Harris hip score, reduction quality, TAD value, fracture healing, and complication rates. The results showed that the PFNA group had a significantly shorter surgical duration [95%CI (-13.70, -3.12)], reduced fluoroscopy time [95%CI (-105.58, -27.32)], and decreased blood loss [95%CI (-25.95, -8.30)] compared to the InterTan group, with statistically significant differences (P<0.001). However, in the postoperative outcome evaluation, there were no significant differences between the two groups in terms of length of hospital stay [95%CI (-1.10, 1.44), P = 0.79], Harris hip score [95%CI (-2.16, 0.50), P = 0.22], reduction quality [95%CI (0.51, 1.86), P = 0.92], TAD value [95%CI (-1.11, 1.30), P = 0.88], and fracture healing [95%CI (-0.01, 0.72), P = 0.06]. Notably, in terms of complication rates, the InterTan group demonstrated a significant advantage compared to the PFNA group. Specifically, the InterTan group had lower complication rates than the PFNA group in Shaft fractures [95%CI (2.39, 12.35), P<0.0001], Varus collapse [95%CI (2.29, 26.53), P = 0.001], Cut out [95%CI (2.82, 13.51),

P<0.0001], screw migration [95%CI (1.70, 17.02), P = 0.004], and hip and thigh pain [95%CI (1.72, 4.93), P<0.0001], with statistically significant differences.

How to achieve high-quality treatment outcomes for intertrochanteric fractures of the femur has always been a central concern for surgeons. In clinical practice, intramedullary fixation has become the dominant surgical approach. However, the choice between PFNA and InterTan has remained a controversial focus when treating such conditions. After thorough analysis, we have found that the postoperative effects of both PFNA and InterTan in the treatment of intertrochanteric fractures of the femur are comparable. Nevertheless, it is noteworthy that PFNA exhibits significant advantages over InterTan in intraoperative indicators, particularly in terms of surgical duration, fluoroscopy time, and blood loss. This is particularly crucial for patients with poor physical conditions who cannot tolerate prolonged surgeries. Despite these similarities, PFNA and InterTan exhibit distinct characteristics in terms of postoperative complications. Although both may experience the same complications, this study has found that InterTan demonstrates a significant advantage over PFNA in reducing the incidence of postoperative complications. Specifically, the incidence of complications such as axial fractures, varus collapse, and screw migration is significantly lower in the InterTan group compared to the PFNA group. This discovery suggests that choosing the InterTan surgical approach may help reduce postoperative risks for patients, thereby enhancing their satisfaction and recovery outcomes. When making a decision on the surgical approach, it is essential to comprehensively consider the patient's overall health status, surgical risks, and expected recovery outcomes. For patients with poor physical conditions who cannot tolerate prolonged surgeries, PFNA may be a more suitable option due to its intraoperative advantages. On the other hand, for patients who can tolerate longer surgeries or face complex fracture situations, InterTan may be a better choice due to its lower incidence of postoperative complications, contributing to improved long-term prognosis. In conclusion, both PFNA and InterTan have their respective advantages and applicable scenarios in the treatment of intertrochanteric fractures of the femur. Surgeons need to weigh the pros and cons based on the specific conditions of the patient and choose the most suitable surgical approach to ensure the best treatment outcome for the patient.

The current meta-analysis still has the following limitations that require further improvement and refinement: Limitations of the Current Meta-Analysis: (1) Insufficient Randomized Controlled Trials: A total of 15 studies were included, but the number of randomized controlled trials was insufficient, resulting in a relatively low level of evidence. (2) Variations in Surgical Operations: There may be slight differences in the specific surgical operations among the different studies, which can potentially affect the surgical outcomes. (3) Heterogeneity in Outcome Indicators: The number of studies included for the same outcome indicator varies widely (ranging from 11 to 3), increasing the heterogeneity among studies. (4) Potential Bias from Patient Informed Consent: Clinical studies adhere to the principle of patient informed consent, which may introduce bias and affect the reliability of the meta-analysis conclusions. (5) Lack of Economic Cost Comparison: The economic cost of the two internal fixation materials was rarely mentioned in the included studies, resulting in the inability to compare this factor. (6) Limitation of Subgroup Analysis: The subgroup analysis in this meta-analysis is limited by the incompleteness and difficulty in acquiring data, preventing a systematic and in-depth exploration of treatment responses among different patient groups. (7) Need for Further Validation: Due to the relatively small sample size and limited number of studies, the stability and reliability of the study results need to be further validated by larger-scale studies. (8) Potential Deviation from Actual Situation: Our conclusions may deviate from the actual situation to a certain extent, necessitating further follow-up studies. We look forward to the emergence of more literature and studies in the future to reduce bias and derive more authentic and

reliable conclusions. In summary, while the current meta-analysis provides valuable insights into the treatment of intertrochanteric fractures, it is important to recognize the limitations discussed above. The small number of randomized controlled trials, variations in surgical operations, heterogeneity in outcome indicators, potential bias from patient informed consent, lack of economic cost comparison, and the need for further validation and follow-up studies indicate that future research should aim to address these issues and refine the evidence base. This will enable us to derive more authentic and reliable conclusions that can better guide clinical practice.

## Supporting information

**S1 Checklist. PRISMA 2020 checklist.**
(DOCX)

**S1 Table. Characteristics of included literature studies.**
(DOCX)

## Author Contributions

**Data curation:** Chang-sheng Liao, Peng-fei Han.

**Formal analysis:** Peng-fei Han.

**Methodology:** Peng-fei Han.

**Project administration:** Xi-yong Li.

**Software:** Chang-sheng Liao.

**Supervision:** Xi-yong Li.

**Visualization:** Chang-sheng Liao.

**Writing – original draft:** Chang-sheng Liao, Fang-zheng He.

**Writing – review & editing:** Chang-sheng Liao, Fang-zheng He.

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
