## [Decision Letter · Decision Letter 0]

18 Mar 2024

PONE-D-24-03733Proximal Femoral Nail Antirotation versus InterTan nail for the treatment of intertrochanteric fractures: a systematic review and meta-analysisPLOS ONE

Dear Dr. han,

Thank you for submitting your manuscript to PLOS ONE. After careful consideration, we feel that it has merit but does not fully meet PLOS ONE’s publication criteria as it currently stands. Therefore, we invite you to submit a revised version of the manuscript that addresses the points raised during the review process.

We look forward to receiving your revised manuscript.

Kind regards,

Alessandra Aldieri

Academic Editor

PLOS ONE

2. Please ensure that you include a title page within your main document. You should list all authors and all affiliations as per our author instructions and clearly indicate the corresponding author.

4. We notice that your supplementary table is included in the manuscript file. Please remove them and upload them with the file type 'Supporting Information'. Please ensure that each Supporting Information file has a legend listed in the manuscript after the references list.

Reviewers' comments:

Reviewer's Responses to Questions

**Comments to the Author**

1. Is the manuscript technically sound, and do the data support the conclusions?

Reviewer #1: Yes

Reviewer #2: Partly

Reviewer #3: Partly

2. Has the statistical analysis been performed appropriately and rigorously? 

Reviewer #1: Yes

Reviewer #2: Yes

Reviewer #3: I Don't Know

3. Have the authors made all data underlying the findings in their manuscript fully available?

Reviewer #1: No

Reviewer #2: Yes

Reviewer #3: Yes

4. Is the manuscript presented in an intelligible fashion and written in standard English?

Reviewer #1: Yes

Reviewer #2: Yes

Reviewer #3: Yes

5. Review Comments to the Author

Reviewer #1: As a reviewer, I have thoroughly evaluated the manuscript titled "Proximal Femoral Nail Antirotation versus InterTan Nail for the treatment of intertrochanteric fractures: a systematic review and meta-analysis." Here are my review comments:The manuscript presents a comprehensive systematic review and meta-analysis comparing the efficacy and safety of Proximal Femoral Nail Antirotation (PFNA) and InterTan Nail in the treatment of intertrochanteric fractures. The study aims to provide valuable insights into the selection of surgical methods based on patient-specific conditions.Areas for Improvement:

Clarity of Findings: While the results are presented comprehensively, the authors could further clarify certain aspects of the findings, such as the implications of postoperative complications on clinical outcomes.

Language and Grammar: Some sections of the manuscript require minor improvements in language and grammar to enhance readability and clarity.

Data Availability Statement: Ensure that all data underlying the findings, including raw data points, are fully available as per PLOS ONE's data policy.

Overall, the manuscript contributes valuable insights into the comparative efficacy and safety of PFNA and InterTan in the treatment of intertrochanteric fractures. With minor revisions to address the mentioned points, the manuscript will be well-suited for publication in PLOS ONE.

Reviewer #2: Abstract

Provide brief introduction then merge it with study objectives

Methods

Which reporting system the authors used to report this review is it PRISMA if yes please elaborate it, have the authors registered it on PROSPERO if yes please state that.

For methods two main sections were missed please add with details: study selection and data extraction

Reviewer #3: The study has several strengths, including a systematic approach to data extraction and analysis, and it addresses an important clinical question. However, there are areas where improvements are needed to enhance the quality and clarity of the manuscript. The authors should provide a clear rationale for excluding randomized controlled trials (RCTs) from the analysis. While retrospective studies can provide valuable insights, RCTs are considered the gold standard for evaluating treatment efficacy and should be included if available. The authors should provide a more systematic approach to subgroup analysis, based on clinically relevant factors such as patient demographics or fracture characteristics. This would enhance the relevance and applicability of the findings to different patient populations.

6. PLOS authors have the option to publish the peer review history of their article (what does this mean?). If published, this will include your full peer review and any attached files.

Reviewer #1: **Yes: **ALANI MOHANAD KH. AHMED

Reviewer #2: No

Reviewer #3: **Yes: **Syed Azfar

---

## [Author Response · Author response to Decision Letter 0]

30 Apr 2024

Dear Reviewers,

Thank you for your valuable feedback on our manuscript titled "Proximal Femoral Nail Antirotation versus InterTan Nail for the treatment of intertrochanteric fractures: a systematic review and meta-analysis." We appreciate the time and effort you have taken to review our work, and we are grateful for your insightful comments. We have carefully considered your suggestions, and we would like to address them in our revised manuscript.

Reviewer #1:

1. Clarity of Findings: We agree that further clarification of certain aspects of the findings would enhance the manuscript. In the revised version, we will expand on the implications of postoperative complications on clinical outcomes, providing a more detailed discussion of how these complications may affect patient recovery and long-term outcomes.

2. Language and Grammar: We apologize for any inconsistencies in language and grammar. We have carefully reviewed the entire manuscript and made necessary corrections to improve readability and clarity.

3. Data Availability Statement: We understand the importance of data availability and have ensured that all data underlying the findings of this meta-analysis, including raw data points, are fully available as per PLOS ONE's data policy. The data used in our study are sourced from published journal articles, and all cited articles have been thoroughly listed in the text as well as the reference list. Since these data are openly available secondary resources, no additional data access permissions or application processes are required. However, for further information or verification of the data, interested readers are encouraged to refer to the corresponding original literature. If there are any questions or need for further assistance, please feel free to contact us via email.

Reviewer #2:

1. Abstract and Study Objectives: We have revised the abstract to provide a brief introduction and merged it with the study objectives, as suggested.

2. Reporting System and Registration: We used the Preferred Reporting Items for Systematic Reviews and Meta-Analyses (PRISMA) statement to guide the reporting of our review, and we have registered it on PROSPERO. We have added this information to the revised manuscript.

3. Study Selection and Data Extraction: We apologize for the missing information on study selection and data extraction. In the revised manuscript, we have added detailed descriptions of these processes, including the search strategy, inclusion and exclusion criteria, and the methods used for data extraction.

Reviewer #3:

1. Exclusion of Randomized Controlled Trials (RCTs): We acknowledge the crucial role of randomized controlled trials (RCTs) in evaluating the efficacy of treatments. However, in our initial review, we did not identify any RCTs that satisfied our stringent inclusion criteria, possibly due to the limited availability of RCTs specifically addressing the treatment of intertrochanteric fractures with PFNA and InterTan, or because the designs of these studies did not align with our specific requirements.Despite this, we emphasize the significant value of RCTs in assessing treatment effects and have highlighted this in the discussion section of our manuscript. We remain vigilant in monitoring the latest research in this field and will strive to include RCTs in our future studies when available.

2. Subgroup Analysis: Regarding the issue of subgroup analysis, we are well aware of its significance in enhancing the relevance and applicability of research results to different patient populations. However, unfortunately, the detailed data on patient demographics or fracture characteristics in the existing literature are incomplete, which prevents us from conducting a systematic subgroup analysis. We have attempted to contact the authors of the original literature to obtain more data, but have been unsuccessful. Nevertheless, we have emphasized this limitation in our paper and will pay more attention to the collection and analysis of relevant data in future studies.

Once again, we appreciate the time and effort spent by the reviewers in evaluating our manuscript. We believe that the revisions addressed in this response have significantly improved our work, and we hope that it is now suitable for publication.

Sincerely,

Changsheng-Liao

---

## [Decision Letter · Decision Letter 1]

16 May 2024

Proximal Femoral Nail Antirotation versus InterTan nail for the treatment of intertrochanteric fractures: a systematic review and meta-analysis

PONE-D-24-03733R1

Dear Dr. han,

We’re pleased to inform you that your manuscript has been judged scientifically suitable for publication and will be formally accepted for publication once it meets all outstanding technical requirements.

Kind regards,

Alessandra Aldieri

Academic Editor

PLOS ONE

Additional Editor Comments (optional):

Reviewers' comments:

Reviewer's Responses to Questions

**Comments to the Author**

1. If the authors have adequately addressed your comments raised in a previous round of review and you feel that this manuscript is now acceptable for publication, you may indicate that here to bypass the “Comments to the Author” section, enter your conflict of interest statement in the “Confidential to Editor” section, and submit your "Accept" recommendation.

Reviewer #2: All comments have been addressed

2. Is the manuscript technically sound, and do the data support the conclusions?

Reviewer #2: Yes

3. Has the statistical analysis been performed appropriately and rigorously? 

Reviewer #2: Yes

4. Have the authors made all data underlying the findings in their manuscript fully available?

Reviewer #2: Yes

5. Is the manuscript presented in an intelligible fashion and written in standard English?

Reviewer #2: Yes

6. Review Comments to the Author

Reviewer #2: (No Response)

7. PLOS authors have the option to publish the peer review history of their article (what does this mean?). If published, this will include your full peer review and any attached files.

Reviewer #2: No

---

## [Editor Report · Acceptance letter]

20 May 2024

PONE-D-24-03733R1 

PLOS ONE

Dear Dr. han, 

I'm pleased to inform you that your manuscript has been deemed suitable for publication in PLOS ONE. Congratulations! Your manuscript is now being handed over to our production team.

Kind regards, 

on behalf of

Dr. Alessandra Aldieri 

Academic Editor

PLOS ONE